Replication

cognition/behaviour

conservation, wildlife, hybridization, brain size, skull, cranial capacity

**Author for correspondence:**
Raffaela Lesch
e-mail: raffaela.lesch@vetmetuni.ac.at

# Cranial volume and palate length of cats, *Felis* spp., under domestication, hybridization and in wild populations

Raffaela Lesch[1,3], Andrew C. Kitchener[2], Georg Hantke[2], Kurt Kotrschal[1] and W. Tecumseh Fitch[1]

[1]Department of Behavioral and Cognitive Biology, Faculty of Life Sciences, University of Vienna, Vienna, Austria
[2]Department Natural Sciences, National Museums Scotland, Edinburgh, UK
[3]Institute of Animal Welfare Science, University for Veterinary Medicine, Vienna, Austria

RL, 0000-0001-7151-252X; ACK, 0000-0003-2594-0827; WTF, 0000-0003-1830-0928

Reduced brain size, compared with wild individuals, is argued to be a key characteristic of domesticated mammal species, and often cited as a key component of a putative 'domestication syndrome'. However, brain size comparisons are often based on old, inaccessible literature and in some cases drew comparisons between domestic animals and wild species that are no longer thought to represent the true progenitor species of the domestic species in question. Here we replicate studies on cranial volumes in domestic cats that were published in the 1960s and 1970s, comparing wildcats, domestic cats and their hybrids. Our data indicate that domestic cats indeed, have smaller cranial volumes (implying smaller brains) relative to both European wildcats (*Felis silvestris*) and the wild ancestors of domestic cats, the African wildcats (*Felis lybica*), verifying older results. We further found that hybrids of domestic cats and European wildcats have cranial volumes that cluster between those of the two parent species. Apart from replicating these studies, we also present new data on palate length in *Felis* cat skulls, showing that domestic cat palates are shorter than those of European wildcats but longer than those of African wildcats. Our data are relevant to current discussions of the causes and consequences of the 'domestication syndrome' in domesticated mammals.

# 1. Background

Understanding differences in morphology in domestic species, their wild ancestors, closely related species and their hybrids are crucial on several fronts. First, in evaluating how domestic animals changed during the extended evolutionary process of domestication, and second, in supporting conservation efforts of wild species threatened by hybridization with domestic animals. Here we investigate the impact of domestication and hybridization on cat cranial (i.e. brain) volume by replicating results presented by Hemmer [1] and Schauenberg [2], and by discussing new results on palate length.

Over the years a vast amount of literature has accumulated, describing various characteristics of domestic mammals (e.g. curly tails, floppy ears, white patches, shorter muzzles; [3,4]). In particular, changes to cranial volume have been well documented across species, including sheep, rabbits, dogs and many more [3,5,6]. A new hypothesis may offer a unifying explanation for these typical traits of domestic mammals. The neural crest cell hypothesis describes how selection for tameness in the domestication of animals may have caused a downregulation in the migration and proliferation of neural crest cells, leading to decreased excitability and fear (tameness). However, this downregulation may also cause correlated changes to morphology, stress response and brain size [7,8]. Although this hypothesis has found a lot of research support, others are critical of it [9]. Lord and colleagues pointed out valid criticism regarding the long-term experiment to domesticate silver foxes, *Vulpes vulpes*, by Belyaev: first, the farmed foxes in Belyaev's domestication experiment were not truly wild but had been bred in captivity since the nineteenth century, and second, morphological comparisons are often problematic, if drawn between specific domestic animal breeds (in this case a variant of the widespread red fox) and/or wild species, which do not represent the true ancestor [10,11]. In the light of the continuing debate about the neural crest cell hypothesis, re-evaluation and replication of Hemmer's [1] and Schauenberg's [2] results are relevant.

Decreased brain size is a commonly described phenomenon in domestic mammals in comparison with their wild ancestors. In cats, a 25% reduction of cranial volume was reported between European wildcats, *Felis silvestris*, and domestic cats, *F. catus* [12,13]. However, work critical of this comparison pointed out that the European wildcat might not be the original ancestor of the domestic cat [1]. An earlier study by Klatt [14] compared cranial volumes of *Felis maniculata* (today *Felis lybica*) with those of house cats and discussed effects of feralization. More recent genetic evidence has confirmed that the African wildcat (*F. lybica*), more specifically the subspecies *F. lybica lybica*, is the ancestor to today's domestic cats [15,16].

Both Hemmer [1] and Schauenberg [2] recognized that the European wildcat was not the ancestral species to domestic cats. Schauenberg [2] formulated the cranial index, i.e. greatest length of skull divided by cranial volume, which clearly separated the skulls of European wildcats from those of domestic cats, whereby domestic cats have smaller brains (cranial index > 2.75) compared with those of European wildcats (cranial index < 2.75). Hemmer set out to compare the cranial volumes of domestic cats with those of European wildcats, various subspecies of the African wildcat and feral domestic cats. His data showed that European wildcats have the largest cranial volumes and domestic cats the smallest. Crucially, all subspecies of African wildcat (*F. lybica*) had cranial volumes smaller than European wildcats, yet bigger than domestic cats. Therefore, Hemmer inferred that domestication had an impact on cat cranial volume, but that effect was smaller than previously reported. To our knowledge no further work exploring this issue has been published, except Groves [17], who presented data on cranial indices of Sardinian wildcats (*F. lybica*) to demonstrate that they had larger cranial volumes than those of domestic cats and to support their recognition as *F. lybica*. It is important to note these differences in cranial volume between different domestic cat and wildcat taxa as part of the wider discussion as to whether domestic cats are truly domesticated (see Discussion).

In this study, we set out to replicate studies of cranial volume in wildcats and domestic cats as presented in Hemmer [1] and Schauenberg [2]. We hypothesize, in line with Hemmer's and Schauenberg's findings, that domestic cats have the smallest cranial volumes, European wildcats have the largest cranial volumes, African wildcats have intermediate cranial volumes and that hybrid cats (*F. silvestris* × *F. catus*) have cranial volumes that lie between those of the two parent species. We also present new data on palate length, relevant both to Hemmer's suggestion and to the neural crest cell hypothesis, whereby we hypothesize that domestication should lead to a reduction in snout dimensions, and thus palate length, in domestic cats [18].

**Table 1.** Overview of the wildcat taxonomy updated from Hemmer [1] and Schauenberg [2], following Kitchener *et al.* [19].

| *Felis* taxonomy | | |
|---|---|---|
| (sub)species names in Hemmer [1] and Schauenberg [2] | updated (sub)species classification [19] | common name |
| *Felis silvestris silvestris* | *Felis silvestris* | European wildcat |
| *Felis silvestris f. catus* | *Felis catus* | domestic cat |
| *Felis silvestris lybica* | *Felis l. lybica* | North African wildcat |
| *Felis silvestris ornata* | *Felis l. ornata* | Asian wildcat |
| *Felis silvestris* sspp. (referring to African wildcat subspecies) | *Felis l. lybica/cafra* | North and South African wildcats |
| *Felis silvestris f. catus* (feral) | *Felis catus* | feral domestic cat |

# 2. Methods

## 2.1. Cat species classification

Cat species and subspecies names and classifications have changed over the years. We used the classification of Kitchener *et al.* [19] for the cat taxa used in this study from the genus *Felis*, including the European wildcat (*F. silvestris*), the African wildcat (*F. lybica*), the domestic cat (*F. catus*) and *F. silvestris* × *F. catus* hybrids from Scotland. Following Kitchener *et al.* [19], we view *F. lybica lybica*, *F. lybica ornata* and *F. lybica cafra* as subspecies of the African wildcat (*F. lybica*); the two subspecies of the European wildcat (*F. silvestris*) are *F. silvestris silvestris* and *F. silvestris caucasica*.

Our dataset comprised 19 *F. lybica* cats split into the following nominal subspecies (as given on museum labels): two were identified as *F. l. lybica* (as *F. haussa*), 12 as *F. l. ornata*, one as *F. l. cafra* and four as *F. l. gordoni*. Both *F. haussa* and *F. l. gordoni* are now included in *F. l. lybica* [19]. Our dataset also included 20 *F. silvestris* of the following nominal subspecies: 19 *F. s. grampia* and one *F. s. silvestris*, but the former is now included in the nominate subspecies. Finally, our data further included 28 domestic cats (*F. catus*) and 36 *F. catus* × *F. silvestris* hybrids from Scotland.

## 2.2. Schauenberg and Hemmer data and digitization

Schauenberg [2] measured cranial volume in two species: *F. silvestris* and *F. catus*; Hemmer [1] measured cranial volume in three species: *F. silvestris*, *F. catus* (feral and house cats) and *F. lybica*. It is unclear if the feral cats Hemmer presented in his study were indeed feral domestic cats or hybrids of European wildcats and domestic cats. The majority of Hemmer's feral domestic cat data has its origin in Klatt [14], who mentioned the possibility of these individuals being European wildcat/domestic hybrids. Therefore, we used skulls of known European wildcat × domestic cat hybrids (as determined by morphological and genetic criteria [20,21]) in our replication study instead of feral domesticated cat skulls.

Schauenberg's [2] cranial index is the ratio between greatest skull length and cranial volume, whereas Hemmer [1] used both basal skull length (prosthion to basion, see below) and greatest skull length. The results regarding the respective species comparisons showed identical patterns for greatest and basal skull length. Since the most complete comparison across species was presented over basal skull length by Hemmer, we decided to use basal skull measurements combined with cranial volume measurements for our own data collection in this replication study.

To allow for a direct comparison of our results with those published previously, we used the package *digitize* [22] to digitize the data visually presented in Schauenberg's [2] fig. 2 and Hemmer's [1] figs. 1 and 2. We updated the taxonomic classifications according to Kitchener *et al.* [19] as described above and provide an overview of the classification system in table 1.

## 2.3. Original cat skull measurements

We used skulls of *F. silvestris*, *F. lybica*, *F. catus* and *Felis catus* × *F. silvestris* hybrids for our study, which allowed us to compare domestic cats with their known ancestor *F. lybica*, *F. silvestris* with which they are

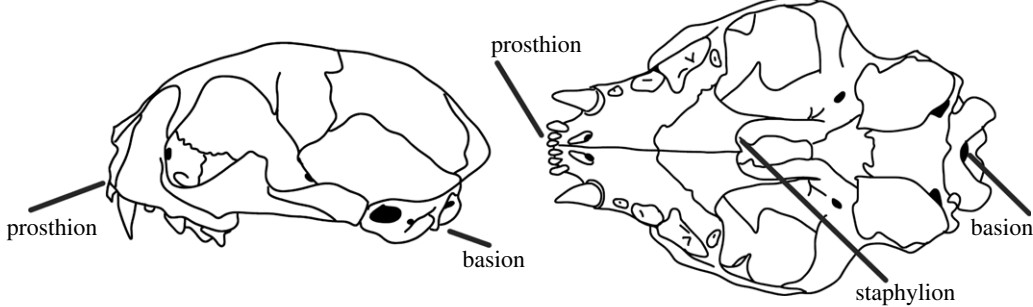

**Figure 1.** Lateral and ventral views of a cat skull indicating the landmarks used for measurements of palate length and basal skull length. Basal skull length was measured from prosthion to basion, and palate length was measured from prosthion to staphylion.

sympatric in Central and Northern Europe, as well as *F. catus × F. silvestris* hybrids. The hybrids (and European wildcats with unclear species identification) used in our study were classified where possible by genetic analysis in Senn *et al.* [21] and morphological analyses of Kitchener *et al.* [20]. In brief, genetic analysis involved a panel of 35 SNPs that had been developed to distinguish wildcats from domestic cats. Cats were identified as wildcat with a LBQ score of 0.75 [21]. Morphological analysis was based on seven key pelage characters with a minimum score of 19 out of 21 for wildcats [20]. In addition, five skull characters were scored [20]. Most of the hybrids originated from the hybrid swarm in Scotland and show varying levels of introgression between the parent species based on genetic and morphological data. Where there was a discrepancy between genetic and morphological evaluations, the default was to record these cats as hybrids. Where genetic data were not available, morphological characters from skins and skulls were used to classify wildcats, domestic cats and their hybrids.

We took three measurements on all cat skulls: Cranial volume, palate length and basal skull length. We measured the cranial volume by closing all openings of the skull except for the foramen magnum with clay and filled the cranium with 1 mm diameter glass beads to the edges of the foramen magnum. This methodology is well established in comparative research and has previously been used in cats [23,24]. We weighed the glass beads that filled the cranium on a balance (Mettler PM4600; Mettler PJ400). To convert the glass bead weight measurements into volumes, we established a conversion factor of 10 ml of glass beads weighing 15.37 g prior to measuring the cranial volumes. Subsequently, we were able to estimate cranial volumes from the weights of the beads.

We measured palate length from the rostral-most premaxilla (prosthion) to the deepest indent of the palatine (staphylion) (figure 1), and basal length of the skull from the tip of the premaxilla to the basion (rostral-most medial indentation of the foramen magnum; figure 1). Both palate length and basal skull length were measured with digital callipers (accurate to 0.01 mm) and measurements were rounded to one decimal point prior to statistical analysis. We measured a total of 103 skulls in the collections of National Museums Scotland. We had to exclude one cat skull from our 'domestic' data group after data collection since its identification number came with conflicting information regarding species identification, leaving us with a total of 102 skulls (27 *F. catus*, 36 *F. silvestris × F. catus* hybrids, 19 *F. lybica*, 20 *F. silvestris*).

## 2.4. Analysis

For our original data, we created linear models with the package *lme4* for cranial volume and palate length [25]. All models included skull length in both the null and full models; species ID was added only to the full models. All full models were compared with their null models, checked for overdispersion, stability, collinearity and residual distribution. The model for palate length indicated an influential observation causing model instability. On closer inspection, this data point was revealed to be an error in recording basal skull length for this individual. Based on digital photos taken from the individual specimen, the value should probably have read 83.8 mm instead of the recorded 63.8 mm. Since equivalent accuracy in measurements from images in comparison with calliper measurements is not guaranteed, we re-ran statistics on both datasets including (see electronic supplementary material) and excluding this individual measurement. Removing the specimen with the measurement error resulted in both the palate length and cranial volume model to be stable and to fulfil all model assumptions.

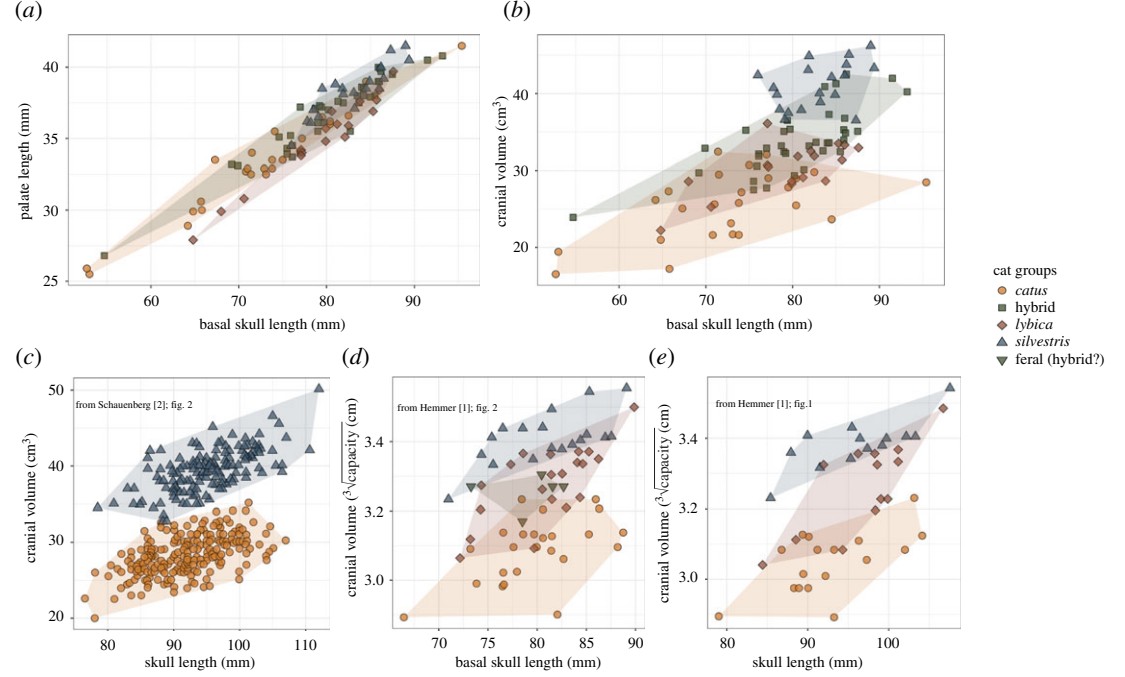

**Figure 2.** Palate length and cranial volume of *Felis* cat species. Domestic cats are represented by orange dots, *F. catus* × *F. silvestris* hybrids by olive squares, *F. lybica* by purple diamonds and *F. silvestris* by blue triangles. (*a*) Palate length of all four groups in mm over basal skull length in mm (our data). (*b*) Cranial volume in cm³ of all four groups over basal skull length in mm (our data). (*c*) Cranial volume data of *F. catus* and *F. silvestris* cats in cm³ over the total skull length from [2]. (*d*) Cube root of cranial volume data for all four groups in cm from [1]. (*e*) Cube root of cranial volume data for *F. catus*, *F. lybica* and *F. silvestris* cats in cm from [1].

## 3. Results

Our data support our hypothesis, as well as the results of Hemmer [1] and Schauenberg [2]. We found that domestic cats have the smallest cranial volume compared with *F. silvestris*, *F. lybica* and *F. silvestris* × *F. catus* hybrids (figure 2*b*). *Felis silvestris* cranial volumes were the largest (estimate ± standard error: 12 ± 1), and *F. lybica* cranial volumes were larger (estimate ± standard error: 2.5 ± 1) than those of domestic cats, but smaller than those of *F. silvestris*. Hybrid cats' (*F. silvestris* × *F. catus*) cranial volumes cluster between the two parent species (estimate ± standard error: 5.6 ± 0.9). While cranial volume was dependent on individuals' size, the relationship between the two is clearly species-specific (table 2).

Our data are less clear with regard to the hypothesis of a reduction in snout length in domestic cats. Contrary to our hypothesis the results indicate that domestic cats have longer palates compared with their ancestral species *F. lybica* (estimate ± standard error: −0.8 ± 0.3). *Felis silvestris* (estimate ± standard error: 1 ± 0.3) has longer palates than those of domestic cats, which have similar palate lengths to those of hybrid cats. While we did find statistically significant differences between species, individual body size appears to be the strongest correlate of palate length (figure 2*a* and table 3). Our full/null model comparison indicated the full model (including the species groups) to be significantly better than the null model, but the adjusted-*R*-squared value only improved marginally from the null to the full model (from 0.91 to 0.93); however, in our cranial volume models this value improved from 0.45 to 0.78 (tables 1 and 2).

## 4. Discussion

In this paper, we replicated results regarding cranial volumes in domestic cats and wildcats first presented by Schauenberg [2] and Hemmer [1]. Our data support their findings that domestic cats have significantly smaller cranial volumes than those of both European wildcats (*F. silvestris*) and African wildcats (*F. lybica*). We further found that hybrids of domestic cats and European wildcats have cranial volumes that cluster between those of the two parent species. We also presented new results on palate length, a proxy for snout length, among domestic cats, European wildcats, African

**Table 2.** Null/full model comparison and summary of the model for cranial volume. Domestic cats are included in the intercept. Residual standard error: 3.032 on 92 degrees of freedom. Multiple $R$-squared: 0.796, adjusted $R$-squared: 0.7871 $F$-statistic: 89.74 on 4 and 92 d.f., $p$-value: $<2.2 \times 10^{-16}$

| cranial volume | | | | |
|---|---|---|---|---|
| | Pr(>Chi) | | | |
| full/null comparison | $<2.2 \times 10^{-16}$ | | | |
| | estimate | s.e. | $t$ value | Pr(>\|t\|) |
| intercept[a] | 0.30317 | 3.28866 | 0.092 | 0.9268 |
| basal skull length | 0.3464 | 0.04445 | 7.792 | $9.71 \times 10^{-12}$ |
| F. catus × F. silvestris hybrid | 5.6224 | 0.86044 | 6.534 | $3.49 \times 10^{-9}$ |
| F. lybica | 2.50871 | 0.9981 | 2.513 | 0.0137 |
| F. silvestris | 12.00224 | 1.02669 | 11.69 | $<2 \times 10^{-16}$ |

[a]Intercept includes domestic cats.

**Table 3.** Null/full model comparison and summary of the model for palate length. Domestic cats are included in the intercept. Residual standard error: 0.844 on 95 degrees of freedom. Multiple $R$-squared: 0.9401, adjusted $R$-squared: 0.9376. $F$-statistic: 373 on 4 and 95 d.f., $p$-value: $<2.2 \times 10^{-16}$

| palate length | | | | |
|---|---|---|---|---|
| | Pr(>Chi) | | | |
| full/null comparison | $3.44 \times 10^{-9}$ | | | |
| | estimate | s.e. | $t$-value | Pr(>\|t\|) |
| intercept[a] | 4.26124 | 0.90774 | 4.694 | $8.99 \times 10^{-6}$ |
| basal skull length | 0.3994 | 0.01224 | 32.626 | $<2 \times 10^{-16}$ |
| F. catus × F. silvestris hybrid | 0.36457 | 0.23586 | 1.546 | 0.125506 |
| F. lybica | −0.77552 | 0.26937 | −2.879 | 0.004928 |
| F. silvestris | 0.96403 | 0.28224 | 3.416 | 0.000938 |

[a]Intercept includes domestic cats.

wildcats and hybrids (*F. silvestris* × *F. catus*). Although we found statistically significant differences among species, individual body size seems to be the main factor driving palate length.

Two characteristics often used to describe changes to domestic animals are a reduction in brain size and snout length. Wilkins *et al.* [7,26] suggested that a neural crest cell deficiency, caused by selection for tameness, is the underlying factor responsible for these characteristics. While a growing body of research (e.g. red junglefowl *Gallus gallus* [27], village dogs *Canis familiaris* [28] and foxes *Vulpes vulpes* [29]) is in line with the neural crest cell hypothesis, the hypothesis has been challenged by others [10,11,30]. Our results on cranial volume reduction in domestic cats are in line with the neural crest cell hypothesis and previous reports [1,31], but we did not find a reduction in snout length, which fails to uphold the prediction of this hypothesis. Dog data presented by Morey [32,33] also show no clear snout length reduction, but rather a relative increase in palate width. There may be several reasons for this lack of snout shortening: domestication might not have affected snout lengths after all, palate length may not be an appropriate proxy for snout length, or the neural crest cell hypothesis may be incorrect in its proposed effect on snout length during domestication.

Lord *et al.* [10,11] outlined two criticisms of the neural crest cell hypothesis that are potentially relevant in this context. First, they pointed out that Dmitri Belyaev's domesticated fox experiment in Siberia began not with 'wild', but with descendants of foxes bred in captivity in Canada for their fur since the nineteenth century and, therefore, (actively or passively) pre-selected for certain traits, such as tameness. Given the partial reliance of the neural crest cell hypothesis on the Belyaev

data, Lord and colleagues argued that it is impossible to distinguish between these changes having emerged due to selection or genetic drift [10]. This potential criticism has been questioned by Zeder [34] and Trut *et al.* [35], who note that Belyaev and his colleagues were well aware of this fact and that all comparisons, both behavioural and morphological, were done between unselected control foxes (from this previously farmed background) and experimentally tamed foxes.

In addition to potential neural crest reductions, other factors may influence morphological changes in domestic animals. An independent factor potentially explaining reductions in brain size in domestic animals is provided by the expensive-tissue hypothesis, which explores evolutionary trade-offs between brain size and other energetically costly tissues [36]. The expensive-tissue hypothesis was originally introduced to explain brain size variability in primates with respect to a trade-off between brain volume and gut size [36]. Since then research on guppies (*Poecilia reticulata*) further supported the idea that brain size is negatively correlated with the size of the gut [37,38]. Further comparative experiments with domestic chicken *Gallus domesticus* and wild junglefowl also support the idea that a trade-off between brain size and other costly systems (e.g. reproduction) is relevant during the domestication process [39]. Therefore, brain size in domestic species might not (only) be affected by e.g. neural crest cell reduction but also by a trade-off between the relative importance of the energetic needs of the brain and other organ systems, such as the gut and/or reproductive system.

Another hypothesis with potential relevance to morphological changes in domestication research is the thyroid hormone hypothesis; this hypothesis was proposed by Crockford [40] and was named and further discussed by Wilkins [8]. This hypothesis states that domestication might have caused timing shifts in development, which potentially affect the concentration of the thyroid hormones during development. Aside from this possible connection between domestication and thyroid hormone concentration, thyroid hormones are essential in the development of craniofacial structures, e.g. deficiencies cause delayed ossification [41]. This potential mechanistic connection between thyroid hormones and craniofacial structures makes this hypothesis an additional candidate for understanding changes to cranial volume and palate length during domestication.

A potential criticism of the use of cat data in domestication research is the oft-mentioned claim that cats are not truly domesticated animals or are only 'semi-domesticated' (cf. [42]). We do not think that this claim is accurate, despite the fact that the cat's path to domestication is often colloquially viewed and portrayed as only beneficial to cats and not humans. Cats might not have been as 'useful' to humans as dogs or horses have been, but their usefulness in keeping grain harvests safe from rodents is commonly cited as a major driver in their domestication [43], and scavenging opportunities at middens may have been as important in bringing wildcats close to humans [44]. Even if wildcats were attracted to human environments because of the ready availability of food, according to Zeder [45] this conforms to the 'commensal pathway' to domestication, probably also relevant in early dog domestication. Furthermore, there can be little doubt that humans have selected for docility in cats, and the fact that (until recently) there has been little further selection for cooperation with humans, as is the case for dogs, or for meat or milk production, as in ungulates, in fact makes cats well-suited to this topic of research [42].

When discussing morphological data in the field of domestication research, it is also highly relevant to integrate the aspect of feralization, which is often viewed as the counter process to domestication [46]. Hemmer [1] provided cranial volumes from domestic pet cats, feral cats (both *F. catus*), *F. silvestris* and *F. lybica*. Hemmer's data for feral cats originated from Klatt [14], who mentioned the possibility of these individuals being hybrid offspring of *F. silvestris* and *F. catus*. Regarding brain size and feralization, research on dingoes, which are domestic dogs in Australia that became feral thousands of years ago [47], shows that brain volume did *not* increase again during the process of feralization [48]. Similar results have been reported for feral cats, goats, mink and pigs [49]. Thus, brain volume reduction due to domestication seems to be a permanent change that is not reversed by feralization, even after many generations [46,49–51]. This permanent change also suggests that reduced brain volume may represent an energy-budget optimization as discussed above. In light of these previous studies, our new data regarding cranial volumes of hybrids suggest that if cats do not regain their ancestral cranial volume in the course of feralization, the putatively 'feral' cats measured and presented by Hemmer/Klatt were actually hybrid cats.

Lord and colleagues note that reports of brain size reduction, considered to be a relatively ubiquitous trait in domestic mammals, vary greatly depending on the data sources, and are sometimes based on comparison with 'wild' animals not representative of the true ancestral species [11]. As the current study shows, the selection of the appropriate 'progenitor' group for comparison with domestic variants is often highly challenging and can have crucial effects on the results.

This leads us to another key issue in comparing wild species and domestic animals: we must always acknowledge that we are comparing a now (or recently) living population of wild animals to the domestic form, and not the true ancestral population. This will always be a confounding factor since we rarely have access to the ancient population that produced our domestic animals (although ancient DNA can partially ameliorate this issue for genetic comparisons). Similarly, we should be cautious in including recently developed breeds in such comparisons. The initial process of domestication is an adaption to a new environment and, therefore, heavily influenced by both natural selection and unconscious selection by humans [26], whereas recent domestic animal breeds have mainly been subjected to conscious artificial selection pressures (e.g. consider farm cats versus Persian cats, or village dogs versus chihuahuas). Comparisons between specific domestic animal breeds and wild populations are interesting, but studies of breeds after strong artificial selection pressures may potentially warp our perception regarding the magnitude and/or type of morphological change present in early domestication [28].

An interesting recent example of the effect of specific breeds is presented by Balcarcel *et al.* [52,53] regarding cattle. The authors compare an impressive dataset of various cattle (*Bos taurus*) breeds with the ancestral aurochs (*Bos primigenius*), finding that all domestic cattle have smaller brains compared with their ancestors. However, among domestic cattle breeds, there are differences in brain size depending on breed temperament; bullfighting breeds have larger brain sizes, possibly related to their *decreased* tameness towards humans. While this is an extremely interesting finding, these differences in brain size and skull morphology among cattle breeds may also be a side effect of artificial selection pressures during breed creation and trade-offs between brain and other tissues (e.g. for meat or milk production, see above) rather than resulting specifically from the domestication process, or breed-specific selection.

To summarize, our results provide confirmation of the previously reported reduction of cranial volume in domestic cats compared with their wild ancestors. We further present new data regarding palate length, showing a lack of snout length reduction in domestic cats. Many studies documenting morphological changes foundational to domestication research are more than 40–50 years old, and in need of replication and further study in accordance with current scientific knowledge and standards (see overview across species by [52,53]). The current study is a step in this direction and helps to solidify the database for an increased understanding of domestication and its effects on morphology.

Data accessibility. All data, code and electronic supplementary materials are publicly available on Dryad: https://doi.org/10.5061/dryad.zgmsbccc5. [54]

Authors' contributions. R.L.: conceptualization, data curation, formal analysis, investigation, methodology, project administration, software, visualization, writing—original draft, Writing—review and editing; A.C.K.: conceptualization, data curation, resources, supervision, writing—review and editing; G.H.: data curation, investigation, resources; K.K.: conceptualization, supervision, writing—review and editing; W.T.F.: conceptualization, funding acquisition, methodology, resources, supervision, writing—review and editing. All authors gave final approval for publication and agreed to be held accountable for the work performed therein.

Competing interests. We declare we have no competing interests.

Funding. Austrian Science Fund (FWF) Grant 'Cognition & Communication 2' (#W1262-B29) to W.T.F.

Acknowledgements. We thank Zena Timmons for her support during data collection. We thank the Negaunee Foundation for its generous support of a curatorial preparator, who prepared many of the skulls used in this study. Many of the specimens in this study were collected by Scottish Wildcat Action, which was supported by the National Lottery Heritage Fund. This article received results-blind in-principle acceptance (IPA) at Royal Society Open Science. Following IPA, the accepted Stage 1 version of the manuscript, not including results and discussion, was preregistered on the OSF (https://osf.io/hdsfq). This preregistration was performed after data analysis. All data and supplementary materials are uploaded to Dryad (https://doi.org/10.5061/dryad.zgmsbccc5).

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
