## [Peer Review File · Royal Society Open Science]

Review History

RSOS-210477.R0 (Original submission)

Review form: Reviewer 1

Do you have any ethical concerns with this paper?

No

Have you any concerns about statistical analyses in this paper?

I do not feel qualified to assess the statistics

Recommendation?

Accept with minor revision

Comments to the Author(s)

This is very interesting work, authors properly justified, why they are reanalyzing samples and results that were already published in the past. I have only few comments.

First is, that cats are sometimes considered as not fully domesticated which may affect their response to the domestication syndrome as authors mention in the introduction. Maybe this topic should be considered to discuss in the paper.

Second is my concern about the species determination. It is not clear how it was made. This topic deserve special chapter, where authors clearly describe how did they evaluated the taxonomy. It seems like the situation is complicated and genetic analyses are necessary not only in case of hybrids. Pure species should be also verified. Later in the text authors mention that several skulls were removed from analyses as they were misidentified. Again, how was this identification done?

Review form: Reviewer 2

Do you have any ethical concerns with this paper?

No

Have you any concerns about statistical analyses in this paper?

I do not feel qualified to assess the statistics

Recommendation?

Accept in principle

Comments to the Author(s)

This is a timely and fascinating study.

As the authors point out, reduced brain size is an iconic (and largely untested) feature of the 'domestication syndrome'.

The authors of this paper are using a fantastic dataset to test the earlier findings of a reduced cranial volume amongst domestic cats relative to a number of different *Felis* species.

I have a few suggestions that I think may improve the manuscript.

The 'domestication syndrome' as a concept was challenged last year in this paper:

<https://doi.org/10.1016/j.tree.2019.10.011>)

A rebuttal to this manuscript (<https://doi.org/10.1016/j.tree.2020.08.009>) suggested that, surely, all domestic animals experienced reduced brain size relative to their wild counterparts.

The authors of the original paper pushed back saying, actually, no, the evidence for reduced brain size is highly problematic. <https://doi.org/10.1016/j.tree.2020.10.004>.

Since then, at least two more papers have been published that have suggested that brain size in domestics is in fact reduced:

<https://royalsocietypublishing.org/doi/10.1098/rspb.2021.0813>

<https://academic.oup.com/jmammal/article/102/1/220/6122317>

There seem to be two issues here.

The first is the choice of wild ancestor. Frequently the 'wild progenitor' species to which the domestics are compared has never had anything to do with domestication.

Secondly, many of these datasets use modern domestic animals and thus, it is impossible to say that, even when cranial volume is smaller, whether that was a feature of the early phases of the domestication process or whether skull volumes have been reduced much more recently for any of a number of reasons that have nothing to do with 'domestication' as it is commonly understood. For instance, regarding the recent cattle paper above, the "purpose" for which the cattle were / are bred is dubious since dairy and meat cattle are heavily selected for morphology, which means behaviour may not be directly or even accidentally selected for. And this selection will have been much much later during the acclimatisation of cows to people over the past 12,000 years.

For the next version of this manuscript, I would love to see the authors consider all the different reasons for why skull volumes may be different, and what other comparisons (different geographical populations of the same species, offspring of captive wild animals, etc.) that could be done to contrast skull volumes in non-domesticatory relationships.

Decision letter (RSOS-210477.R0)

Dear Ms Lesch

On behalf of the Editors, I am pleased to inform you that your Manuscript RSOS-210477 entitled "Cranial volume and palate length of cats, *Felis* spp., under domestication, hybridisation and in wild populations" deemed suitable for in-principle acceptance in Royal Society Open Science subject to minor revision in accordance with the referee and editor suggestions. Please find their comments at the end of this email.

The reviewers and handling editors have recommended publication, but also suggest some minor revisions to your manuscript. Therefore, I invite you to respond to the comments and revise your manuscript.

Please you submit the revised version of your manuscript within 7 days (i.e. by the 19-Jun-2021). If you do not think you will be able to meet this date please let me know immediately.

When submitting your revised manuscript, you will be able to respond to the comments made by the referees and upload a file "Response to Referees" in the "File Upload" step. You can use this to document any changes you make to the original manuscript. In order to expedite the processing of the revised manuscript, please be as specific as possible in your response to the referees.

Full author guidelines can be found here <https://royalsocietypublishing.org/rsos/replication-studies#AuthorsGuidance>.

on behalf of Professor Chris Chambers (Registered Reports Editor, Royal Society Open Science)
openscience@royalsociety.org

Associate Editor Comments to Author (Professor Chris Chambers):

Thank you for your patience during this challenging time for reviewers. Two expert reviewers have now assessed the Stage 1 manuscript. As you will see, the reviews are broadly very positive, judging that the primary Stage 1 criteria are met. The reviewers also note some areas of the manuscript that would benefit from additional clarification and justification, especially concerning prior literature and rationale, as well as additional methodological details. Provided the authors are able to respond comprehensively to these points in a revised manuscript, in-principle acceptance should be forthcoming without requiring further in-depth Stage 1 review.

Reviewers' comments to Author:

Reviewer: 1

Comments to the Author(s)

This is very interesting work, authors properly justified, why they are reanalyzing samples and results that were already published in the past. I have only few comments.

First is, that cats are sometimes considered as not fully domesticated which may affect their response to the domestication syndrome as authors mention in the introduction. Maybe this topic should be considered to discuss in the paper.

Second is my concern about the species determination. It is not clear how it was made. This topic deserve special chapter, where authors clearly describe how did they evaluated the taxonomy. It seems like the situation is complicated and genetic analyses are necessary not only in case of hybrids. Pure species should be also verified. Later in the text authors mention that several skulls were removed from analyses as they were misidentified. Again, how was this identification done?

Reviewer: 2

Comments to the Author(s)

This is a timely and fascinating study.

As the authors point out, reduced brain size is an iconic (and largely untested) feature of the 'domestication syndrome'.

The authors of this paper are using a fantastic dataset to test the earlier findings of a reduced cranial volume amongst domestic cats relative to a number of different *Felis* species.

I have a few suggestions that I think may improve the manuscript.

The 'domestication syndrome' as a concept was challenged last year in this paper:

<https://doi.org/10.1016/j.tree.2019.10.011>)

A rebuttal to this manuscript (<https://doi.org/10.1016/j.tree.2020.08.009>) suggested that, surely, all domestic animals experienced reduced brain size relative to their wild counterparts.

The authors of the original paper pushed back saying, actually, no, the evidence for reduced brain size is highly problematic. <https://doi.org/10.1016/j.tree.2020.10.004>.

Since then, at least two more papers have been published that have suggested that brain size in domestics is in fact reduced:

<https://royalsocietypublishing.org/doi/10.1098/rspb.2021.0813>

<https://academic.oup.com/jmammal/article/102/1/220/6122317>

There seem to be two issues here.

The first is the choice of wild ancestor. Frequently the 'wild progenitor' species to which the domestics are compared has never had anything to do with domestication.

Secondly, many of these datasets use modern domestic animals and thus, it is impossible to say that, even when cranial volume is smaller, whether that was a feature of the early phases of the domestication process or whether skull volumes have been reduced much more recently for any of a number of reasons that have nothing to do with 'domestication' as it is commonly understood. For instance, regarding the recent cattle paper above, the "purpose" for which the cattle were / are bred is dubious since dairy and meat cattle are heavily selected for morphology, which means behaviour may not be directly or even accidentally selected for. And this selection will have been much much later during the acclimatisation of cows to people over the past 12,000 years.

For the next version of this manuscript, I would love to see the authors consider all the different reasons for why skull volumes may be different, and what other comparisons (different geographical populations of the same species, offspring of captive wild animals, etc.) that could be done to contrast skull volumes in non-domesticatory relationships.

Author's Response to Decision Letter for (RSOS-210477.R0)

See Appendix A.

Decision letter (RSOS-210477.R1)

Dear Ms Lesch

On behalf of the Editor, I am pleased to inform you that your Manuscript RSOS-210477.R1 entitled "Cranial volume and palate length of cats, *Felis spp.*, under domestication, hybridisation and in wild populations" has been accepted in principle for publication in Royal Society Open Science.

You may now progress to Stage 2 and complete the study as approved.

Please note that you must now register your approved protocol on the Open Science Framework (<https://osf.io/rr>), using the 'Submit your approved Registered Report' option and then the 'Registered Report Protocol Preregistration' option. Please use the Registered Report option even though your article is being accepted as a Stage 1 Replication. Further into the registration process, in the Journal Title field enter 'Royal Society Open Science (Replication article type, Results-Blind track)'. Please note that a time-stamped, independent registration of the protocol is mandatory under journal policy, and manuscripts that do not conform to this requirement cannot be considered at Stage 2. The protocol should be registered unchanged from its current approved state. Please include a URL to the protocol in your Stage 2 manuscript, and because you submitted via the Results-Blind track please note in the manuscript that the pre-registration was performed after data analysis (e.g. 'This article received results-blind in-principle acceptance (IPA) at Royal Society Open Science. Following IPA, the accepted Stage 1 version of the manuscript, not including results and discussion, was preregistered on the OSF (URL). This preregistration was performed after data analysis.')

We would be grateful if you could now update the journal office as to the anticipated completion date of your study.

Following completion of your study, we invite you to resubmit your paper for peer review as a Stage 2 Replication. Please note that your manuscript can still be rejected for publication at Stage 2 if the Editors consider any of the following conditions to be met:

- The Introduction and methods deviated from the approved Stage 1 submission (required).
- The authors' conclusions were not considered justified given the data.

We encourage you to read the complete guidelines for authors concerning Stage 2 submissions at: <https://royalsocietypublishing.org/rsos/replication-studies#AuthorsGuidance>. Please especially note the requirements for data sharing and that withdrawing your manuscript will result in publication of a Withdrawn Registration.

Once again, thank you for submitting your manuscript to Royal Society Open Science and I look forward to receiving your Stage 2 submission. If you have any questions at all, please do not hesitate to get in touch. We look forward to hearing from you shortly with the anticipated submission date for your stage two manuscript.

Kind regards,
Professor Chris Chambers
Royal Society Open Science
openscience@royalsociety.org

Author's Response to Decision Letter for (RSOS-210477.R1)

See Appendix B.

RSOS-210477.R2

Review form: Reviewer 1

Is the manuscript scientifically sound in its present form?

Yes

Is the language acceptable?

Yes

Do you have any ethical concerns with this paper?

No

Have you any concerns about statistical analyses in this paper?

No

Recommendation?

Accept with minor revision

Comments to the Author(s)

The study re-evaluates the cranial differences between domestic and wild cats. The authors confirmed a reduction of brain size in domestic cats, however failed to observe a reduction of snout size in domestic cats. They confront their results with the most recent literature concerning domestication syndromes.

Compared to the previous version, the introduction part contains a little more information and the number of studied animals changed from 103 to 102.

Overall, I consider the study as very important and worth publishing, however, I believe that the manuscript may benefit from restructuralization and reduction of the discussion. Some parts of the discussion are wordy and hard to follow. Some parts would better fit to the introduction, some parts are repetitions of the introduction or results.

Review form: Reviewer 2

Is the manuscript scientifically sound in its present form?

No

Is the language acceptable?

Yes

Do you have any ethical concerns with this paper?

No

Have you any concerns about statistical analyses in this paper?

No

Recommendation?

Major revision

Comments to the Author(s)

This is a timely and important contribution to domestication studies, not least given the recent vibrancy and frequency of papers surrounding domestication studies writ large. This is Greger Larson, I wanted to sign this review, since I am fascinated by this study and, in this case, it made more sense to me not to have this review be anonymous. I have a few comments that I'm hoping will improve this important manuscript.

The abstract of any paper should summarise the entire manuscript. The authors set up the story nicely, but then stop before presenting any results. This is not a movie preview, it is a paragraph that encapsulates the premise, the results, and the conclusions and I would ask the authors to rewrite the abstract to include every aspect of this paper. What did you find? Tell us!

The field is moving quickly and there has been another challenge to the neural crest hypothesis: DOI: 10.1093/genetics/iyab097. Wilkins has replied to this as well, and these latest entries in the debate should be included here. Having said that, I'm not sure how relevant the neural crest hypothesis is, or rather, it's important, but I would like to see the authors reference earlier papers that simply note, suggest, or observe smaller cranial volumes in domestic taxa (relative to their wild counterparts). Some of these references appear in the second paragraph, but they should be moved up since though Wilkins attempted to provide an explanation for the phenomenon, the observations have been around for decades prior to the NCT. This paper tests the observation, not the mechanism for its appearance, and that should be made clear by restricting the introduction.

The authors mention other taxa in the abstract, but these seem to be missing in the introduction. The Lord et al. response in TREE regarding skull size is a good place to find and include some of these, and there are recent papers by Marcelo Sanchez that would fit well here too.

For the paragraph starting on line 65, how did these authors define 'feral' and 'domestic' cats. What criteria did they use, how many samples did they test, and from what geographical locations?

Line 77. What species is the Sardinian wild cat?

Line 107. How was it determined that the hybrids were in fact hybrids? Were they F1 hybrids, or were they thought to have only a small fraction of one lineage in their ancestry, and how was this known? The references on line 117 are great, but as a reader I'd prefer not to have to look these up. Can you provide a summary of how this determination was made here?

Line 120. This paragraph makes it sound like cranial volume was not measured, but later it's clear that it was. Can you clarify that here? And has the glass bead methodology been used before? Can you cite previous papers where this method has been tested and the figures regarding the weight and volume measurements were ascertained?

Line 184. It would be great to include the basic quantitative results here. I know the results are in the table, but it would be great to include them in the prose as well. Mean, standard error, and statistical significance testing between the groups.

Line 225. Did the authors check if body size differences were also correlated with the cranial volume measurements. Could this be allometry?

Line 232. I wouldn't go so far as to say a growing body of evidence "supports" the NCT. It's more like their data could be consistent with it, but that's a completely different association. The review by Dominic Wright points this out and many other things besides.

Line 235. This opening line is unnecessary and I would delete it since the paragraph words perfectly well without it. The authors also only discuss the NCT. There are other mechanisms that have been published, including the thyroid hormone, to explain the domestication syndrome, and it might be worth mentioning them here as well.

Line 265. There is a new paper about the archaeologically determined time frame for the arrival of dingoes by Sue O'Connor that could be cited here. It would give some grounding to the phrase thousands of years ago.

Line 303. The point that Lord was making is that all of the foxes came from the same population that had been through two big bottlenecks before arriving in Russia. The argument in the paper is that there's no way to distinguish the appearance of these traits as a result of either selection or drift, since the population started with a seriously limited degree of genomic variation.

Line 307. To say "Furthermore, the degree to which the Canadian farmed foxes showed domesticated behaviours was overstated by Lord and colleagues." Without any supporting evidence or a citation is an empty accusation. If this was an overstatement, you have to say why you are saying that on the basis of what evidence. Scientific claims need to be supported by evidence, and not simply stated as fact without any empirical or observational evidence.

Line 310. This is not the case. The fixation of these traits could easily result from drift, especially given the winnowing of the genomic variation through successive rounds of selection for tameness. What I'm not understanding is why this is relevant. This study is a test of the differences in cranial volume, not a defence of the NCT as a mechanism to explain the domestication syndrome, which again, the very appearance of which has been challenged by numerous studies cited in this paper (e.g. Hansen Wheat, etc.)

And with regard to that issue, showing a difference in the skull volumes of domestic and wild individuals is fascinating. But it does not pin point when, during the process of domestication, this took place. There is not reason that the drop in skull volume could not have occurred very recently, especially if all the domestic cats are house cats. Just something to consider.

Line 363. The authors refer to the "neural crest/domestication syndrome hypothesis" but this conflates two very different things. The "domestication syndrome" is a suggestion that many different domestic animals possess a suite of similarly divergent characteristics relative to their individual wild ancestors. The NCT assumed the domestication syndrome to be an accurate observation, and is a proposed hypothesis to explain the appearance of those traits. There are hypotheses other than the NCT that have been discussed in the literature (e.g. DOI 10.18699/VJ17.262) as explanations for the domestication syndrome, recent studies have challenged the NCT (the Wright paper above) and Lord et al. challenged the existence of the domestication syndrome itself.

I encourage the authors to clearly distinguish all these terms and debates and to be explicit about what this study is doing, and what can be gleaned from their results.

It's a fantastic paper and with some tweaking and clarifying it'll be an excellent contribution to the literature.

Decision letter (RSOS-210477.R2)

Dear Ms Lesch

On behalf of the Editor, I am pleased to inform you that your Stage 2 Replication submission RSOS-210477.R2 entitled "Cranial volume and palate length of cats, *Felis spp.*, under domestication, hybridisation and in wild populations" has been accepted for publication in Royal Society Open Science subject to minor revision in accordance with the referee suggestions. Please find the referees' comments at the end of this email.

The reviewers and Subject Editor have recommended publication, but also suggest some minor revisions to your manuscript. We invite you to respond to the comments and revise your manuscript. Below the referees' and Editors' comments (where applicable) we provide additional requirements. Final acceptance of your manuscript is dependent on these requirements being met. We provide guidance below to help you prepare your revision.

Please submit your revised manuscript and required files (see below) no later than 7 days from today's (ie 15-Nov-2021) date. Note: the ScholarOne system will 'lock' if submission of the revision is attempted 7 or more days after the deadline. If you do not think you will be able to meet this deadline please contact the editorial office immediately.

Kind regards,
Professor Chris Chambers
Royal Society Open Science
openscience@royalsociety.org

Associate Editor Comments to Author (Professor Chris Chambers):

Associate Editor: 1

Comments to the Author:

The two expert reviewers from Stage 1 kindly returned to evaluate the Stage 2 manuscript, and happily, both are very positive about the completed manuscript. As you will see, most of the comments concern the clarity and cohesion of the Discussion, which is where the bulk of your efforts in revision should be focused. In revising, please also (1) update the Abstract to include a summary of the results and conclusions, and (2) ensure that in responding to the reviewer's comments, no changes to the Introduction and Methods are made other than necessary to ensure clarity and correct any factual errors. You can, however, clarify certain issues raised by Reviewer 2. Provided you are able to respond comprehensively to these reviews, full acceptance should be forthcoming without requiring further in-depth review.

Reviewers' comments to Author:

Reviewer: 1

Comments to the Author(s)

The study re-evaluates the cranial differences between domestic and wild cats. The authors confirmed a reduction of brain size in domestic cats, however failed to observe a reduction of snout size in domestic cats. They confront their results with the most recent literature concerning domestication syndromes.

Compared to the previous version, the introduction part contains a little more information and the number of studied animals changed from 103 to 102.

Overall, I consider the study as very important and worth publishing, however, I believe that the manuscript may benefit from restructuralization and reduction of the discussion. Some parts of the discussion are wordy and hard to follow. Some parts would better fit to the introduction, some parts are repetitions of the introduction or results.

Reviewer: 2

Comments to the Author(s)

This is a timely and important contribution to domestication studies, not least given the recent vibrancy and frequency of papers surrounding domestication studies writ large. This is Greger Larson, I wanted to sign this review, since I am fascinated by this study and, in this case, it made more sense to me not to have this review be anonymous. I have a few comments that I'm hoping will improve this important manuscript.

The abstract of any paper should summarise the entire manuscript. The authors set up the story nicely, but then stop before presenting any results. This is not a movie preview, it is a paragraph that encapsulates the premise, the results, and the conclusions and I would ask the authors to rewrite the abstract to include every aspect of this paper. What did you find? Tell us!

The field is moving quickly and there has been another challenge to the neural crest hypothesis: DOI: 10.1093/genetics/iyab097. Wilkins has replied to this as well, and these latest entries in the debate should be included here. Having said that, I'm not sure how relevant the neural crest hypothesis is, or rather, it's important, but I would like to see the authors reference earlier papers that simply note, suggest, or observe smaller cranial volumes in domestic taxa (relative to their wild counterparts). Some of these references appear in the second paragraph, but they should be moved up since though Wilkins attempted to provide an explanation for the phenomenon, the observations have been around for decades prior to the NCT. This paper tests the observation, not the mechanism for its appearance, and that should be made clear by restricting the introduction.

The authors mention other taxa in the abstract, but these seem to be missing in the introduction. The Lord et al. response in TREE regarding skull size is a good place to find and include some of these, and there are recent papers by Marcelo Sanchez that would fit well here too.

For the paragraph starting on line 65, how did these authors define 'feral' and 'domestic' cats. What criteria did they use, how many samples did they test, and from what geographical locations?

Line 77. What species is the Sardinian wild cat?

Line 107. How was it determined that the hybrids were in fact hybrids? Were they F1 hybrids, or were they thought to have only a small fraction of one lineage in their ancestry, and how was this

known? The references on line 117 are great, but as a reader I'd prefer not to have to look these up. Can you provide a summary of how this determination was made here?

Line 120. This paragraph makes it sound like cranial volume was not measured, but later it's clear that it was. Can you clarify that here? And has the glass bead methodology been used before? Can you cite previous papers where this method has been tested and the figures regarding the weight and volume measurements were ascertained?

Line 184. It would be great to include the basic quantitative results here. I know the results are in the table, but it would be great to include them in the prose as well. Mean, standard error, and statistical significance testing between the groups.

Line 225. Did the authors check if body size differences were also correlated with the cranial volume measurements. Could this be allometry?

Line 232. I wouldn't go so far as to say a growing body of evidence "supports" the NCT. It's more like their data could be consistent with it, but that's a completely different association. The review by Dominic Wright points this out and many other things besides.

Line 235. This opening line is unnecessary and I would delete it since the paragraph words perfectly well without it. The authors also only discuss the NCT. There are other mechanisms that have been published, including the thyroid hormone, to explain the domestication syndrome, and it might be worth mentioning them here as well.

Line 265. There is a new paper about the archaeologically determined time frame for the arrival of dingoes by Sue O'Connor that could be cited here. It would give some grounding to the phrase thousands of years ago.

Line 303. The point that Lord was making is that all of the foxes came from the same population that had been through two big bottlenecks before arriving in Russia. The argument in the paper is that there's no way to distinguish the appearance of these traits as a result of either selection or drift, since the population started with a seriously limited degree of genomic variation.

Line 307. To say "Furthermore, the degree to which the Canadian farmed foxes showed domesticated behaviours was overstated by Lord and colleagues." Without any supporting evidence or a citation is an empty accusation. If this was an overstatement, you have to say why you are saying that on the basis of what evidence. Scientific claims need to be supported by evidence, and not simply stated as fact without any empirical or observational evidence.

Line 310. This is not the case. The fixation of these traits could easily result from drift, especially given the winnowing of the genomic variation through successive rounds of selection for tameness. What I'm not understanding is why this is relevant. This study is a test of the differences in cranial volume, not a defence of the NCT as a mechanism to explain the domestication syndrome, which again, the very appearance of which has been challenged by numerous studies cited in this paper (e.g. Hansen Wheat, etc.)

And with regard to that issue, showing a difference in the skull volumes of domestic and wild individuals is fascinating. But it does not pin point when, during the process of domestication, this took place. There is not reason that the drop in skull volume could not have occurred very recently, especially if all the domestic cats are house cats. Just something to consider.

Line 363. The authors refer to the "neural crest/domestication syndrome hypothesis" but this conflates two very different things. The "domestication syndrome" is a suggestion that many

different domestic animals possess a suite of similarly divergent characteristics relative to their individual wild ancestors. The NCT assumed the domestication syndrome to be an accurate observation, and is a proposed hypothesis to explain the appearance of those traits. There are hypotheses other than the NCT that have been discussed in the literature (e.g. DOI 10.18699/VJ17.262) as explanations for the domestication syndrome, recent studies have challenged the NCT (the Wright paper above) and Lord et al. challenged the existence of the domestication syndrome itself.

I encourage the authors to clearly distinguish all these terms and debates and to be explicit about what this study is doing, and what can be gleaned from their results.

It's a fantastic paper and with some tweaking and clarifying it'll be an excellent contribution to the literature.

===PREPARING YOUR MANUSCRIPT===

one version should clearly identify all the changes that have been made (for instance, in coloured highlight, in bold text, or tracked changes);

===PREPARING YOUR REVISION IN SCHOLARONE===

To revise your manuscript, log into <https://mc.manuscriptcentral.com/rsos> and enter your Author Centre - this may be accessed by clicking on "Author" in the dark toolbar at the top of the

page (just below the journal name). You will find your manuscript listed under "Manuscripts with Decisions". Under "Actions", click on "Create a Revision".

-- Ensure that your data access statement meets the requirements at <https://royalsociety.org/journals/authors/author-guidelines/#data>.

You should ensure that you cite the dataset in your reference list. If you have deposited data etc in the Dryad repository, please only include the 'For publication' link at this stage. You should remove the 'For review' link.

-- If you are requesting an article processing charge waiver, you must select the relevant waiver option (if requesting a discretionary waiver, the form should have been uploaded, see 'File upload' above).

-- If you have uploaded any electronic supplementary (ESM) files, please ensure you follow the guidance at <https://royalsociety.org/journals/authors/author-guidelines/#supplementary-material> to include a suitable title and informative caption. An example of appropriate titling and captioning may be found at https://figshare.com/articles/Table_S2_from_Is_there_a_trade-off_between_peak_performance_and_performance_breadth_across_temperatures_for_aerobic_scope_in_teleost_fishes_/3843624.

At the 'Review & submit' step, you must view the PDF proof of the manuscript before you will be able to submit the revision. Note: if any parts of the electronic submission form have not been

completed, these will be noted by red message boxes - you will need to resolve these errors before you can submit the revision.

Author's Response to Decision Letter for (RSOS-210477.R2)

See Appendix C.

Decision letter (RSOS-210477.R3)

Dear Ms Lesch:

It is a pleasure to accept your Stage 2 Replication entitled "Cranial volume and palate length of cats, *Felis spp.*, under domestication, hybridisation and in wild populations" in its current form for publication in Royal Society Open Science.

on behalf of Professor Chris Chambers (Subject Editor)
openscience@royalsociety.org

Appendix A

author response to reviewers

2.7.2021

RSOS-210477 Cranial volume and palate length of cats, *Felis* spp., under domestication, hybridisation and in wild populations

Reviewer comments are marked in **bold** and the authors response are below in *italic*.

Reviewer: 1

1- This is very interesting work, authors properly justified, why they are reanalyzing samples and results that were already published in the past. I have only few comments.

Thank you!

2- First is, that cats are sometimes considered as not fully domesticated which may affect their response to the domestication syndrome as authors mention in the introduction. Maybe this topic should be considered to discuss in the paper.

We included this view in the introduction (line 81) and will also further explore this point in the discussion:

“It is important to note these differences in cranial volume between different domestic cat and wildcat taxa as part of the wider discussion as to whether domestic cats are truly domesticated (see Discussion).”

3- Second is my concern about the species determination. It is not clear how it was made. This topic deserve special chapter, where authors clearly describe how did they evaluated the taxonomy. It seems like the situation is complicated and genetic analyses are necessary not only in case of hybrids. Pure species should be also verified.

The critical issue is the identification of hybrids, some of which may look superficially like either wildcats or domestic cats. The hybrid cats are from Scotland where the hybridisation between domestic cats and European wildcats is especially worrying. Since genetic analysis on all cats would have been beyond the scope of this paper we used the genetic analysis presented (on exactly the same hybrid cats we present here) by Senn et al. (2019). Any European wildcats where there was doubt regarding their species identification has also been analysed previously by Senn et al. Domestic cats are easily distinguished from wildcats and hybrids on morphology alone, through e.g. pelage coloration and markings, limb bone length. The ornata cats were from zoos and their morphology was characteristic (the fact that their cranial volumes were larger than those of domestic cats show that they weren't hybrids or domestic cats). The southern African wildcats were from a remote area of the Kalahari where there are no domestic cats and again their morphology was consistent with this identification. We added information regarding this issue starting in line 102 and 141:

“The hybrids (and European wildcats with unclear species identification) used in our study were classified as such by genetic analysis in Senn et al. (2019) and morphological analyses of Kitchener et al. (2005).”

“Our data set comprised 19 Felis lybica cats split into the following nominal subspecies (as given on museum labels): 11 were identified as Felis l. lybica, two as F. l. ornata, one as F. l. haussa, one as F. l. cafra and four as F. l. gordonii. Both F. l. haussa and F. l. gordonii are now included in F. l. lybica (Kitchener et al., 2017). Our data set also included 21 F. silvestris of the following nominal subspecies: ...”

4- Later in the text authors mention that several skulls were removed from analyses as they were misidentified. Again, how was this identification done?

We clarified the text regarding the removal of (the 2) specimen from the data set. One individual was excluded after analysis indicated a severe outlier and cross-referencing with our picture data confirmed a typo regarding this individual's data entry (line 173). The second skull was excluded after data collection because the individual identification came with conflicting information regarding species (line 155). Other than these two cats no other cat skulls were excluded from our analysis.

Reviewer: 2

5- This is a timely and fascinating study. As the authors point out, reduced brain size is an iconic (and largely untested) feature of the ‘domestication syndrome’. The authors of this paper are using a fantastic dataset to test the earlier findings of a reduced cranial volume amongst domestic cats relative to a number of different Felis species.

Thank you!

6- I have a few suggestions that I think may improve the manuscript. The ‘domestication syndrome’ as a concept was challenged last year in this paper: <https://doi.org/10.1016/j.tree.2019.10.011>)

We now included a paragraph in the introduction outlining the criticism the neural crest cell received (starting line 46). We discuss this in further detail in the discussion.

“This hypothesis has found a lot of research support, but others are critical of the hypothesis. Lord and colleagues pointed out valid criticism regarding the long-term experiment to domesticate silver foxes, Vulpes vulpes, by Belyaev : First the farmed foxes in Belyaev's domestication experiment were not truly wild, but had been bred in captivity since the nineteenth century, and second morphological comparisons are often problematic if drawn between specific domestic animal breeds (in this case a variant of the widespread red fox) and/or wild species, which do not represent the true ancestor (Lord et al., 2020; Lord, Larson and Karlsson, 2020). In light of the continuing debate about the neural crest cell hypothesis re-evaluation and replication of Hemmer's (1972) and Schauenberg's (1969) results are relevant.”

7- A rebuttal to this manuscript (<https://doi.org/10.1016/j.tree.2020.08.009>) suggested

that, surely, all domestic animals experienced reduced brain size relative to their wild counterparts. The authors of the original paper pushed back saying, actually, no, the evidence for reduced brain size is highly problematic. <https://doi.org/10.1016/j.tree.2020.10.004>. Since then, at least two more papers have been published that have suggested that brain size in domestics is in fact reduced:

<https://royalsocietypublishing.org/doi/10.1098/rspb.2021.0813>

<https://academic.oup.com/jmammal/article/102/1/220/6122317>

There seem to be two issues here:

The first is the choice of wild ancestor. Frequently the ‘wild progenitor’ species to which the domestics are compared has never had anything to do with domestication.

Secondly, many of these datasets use modern domestic animals and thus, it is impossible to say that, even when cranial volume is smaller, whether that was a feature of the early phases of the domestication process or whether skull volumes have been reduced much more recently for any of a number of reasons that have nothing to do with

‘domestication’ as it is commonly understood. For instance, regarding the recent cattle paper above, the “purpose” for which the cattle were / are bred is dubious since dairy and meat cattle are heavily selected for morphology, which means behaviour may not be directly or even accidentally selected for. And this selection will have been much much later during the acclimitisation of cows to people over the past 12,000 years.

We also included this problematic of comparing “wrong” ancestor species and/or domestic animal breeds in our introduction in a cat specific context (starting line 50; manuscript section copied under above answer). A more detailed discussion on this general topic will follow in the discussion. We agree, this is a very important point and we want to thank reviewer 2 for sharing these papers with us. This is highly relevant in discussing our results and we will include these papers in the discussion of our manuscript.

8- For the next version of this manuscript, I would love to see the authors consider all the different reasons for why skull volumes may be different, and what other comparisons (different geographical populations of the same species, offspring of captive wild animals, etc.) that could be done to contrast skull volumes in non-domesticatory relationships.

We now make short mention of on aspect of this in the introduction (starting line 79) but again will delve into details (and all aspects regarding this) in the discussion.

“It is important to note these differences in cranial volume between different domestic cat and wildcat taxa as part of the wider discussion as to whether domestic cats are truly domesticated (see Discussion).”

Appendix B

Coverletter Royal Society Open Science

Vienna, 27.8.2021

Dear Professor Chambers,

We finished the revisions of our manuscript RSOS-210477.R1 "Cranial volume and palate length of cats, *Felis* spp., under domestication, hybridisation and in wild populations" and are ready to submit it to stage 2 of the results-blind track for replication studies.

We followed the instructions provided and preregistered our study at OSF (<https://osf.io/hdsfq>) and uploaded our data and code to be freely accessible on dryad (<https://doi.org/10.5061/dryad.zgmsbccc5>).

The abstract, introduction and methods have not been altered after in principle acceptance in stage 1 with the exception of the correction of two typos. One typo was corrected in the authors affiliation and the second typo (repeated three times) was corrected regarding the number of skulls which was off by a count of one. This typo did not affect or change the analysis or methods used.

We further discuss all points that were raised by the reviewers during stage 1 in the discussion.

Sincerely yours, for the authors,
Raffaella Lesch

Appendix C

author response to reviewers

14.12.2021

RSOS-210477.R2 Cranial volume and palate length of cats, *Felis* spp., under domestication, hybridisation and in wild populations

Reviewer comments are marked in **bold** and the authors response are below in *italic*.

Associate Editor Professor Chris Chambers:

The two expert reviewers from Stage 1 kindly returned to evaluate the Stage 2 manuscript, and happily, both are very positive about the completed manuscript. As you will see, most of the comments concern the clarity and cohesion of the Discussion, which is where the bulk of your efforts in revision should be focused. In revising, please also (1) update the Abstract to include a summary of the results and conclusions, and (2) ensure that in responding to the reviewer's comments, no changes to the Introduction and Methods are made other than necessary to ensure clarity and correct any factual errors. You can, however, clarify certain issues raised by Reviewer 2. Provided you are able to respond comprehensively to these reviews, full acceptance should be forthcoming without requiring further in-depth review.

Dear Professor Chambers, we revised the manuscript according to your points and the reviewers suggestions. The abstract is updated, the introduction and methods section remained the same and were only altered in minor details for clarification (marked in the manuscript) in accordance with reviewer comments. We also restructured the discussion and reduced it to the more relevant aspects of this paper.

Reviewer: 1

The study re-evaluates the cranial differences between domestic and wild cats. The authors confirmed a reduction of brain size in domestic cats, however failed to observe a reduction of snout size in domestic cats. They confront their results with the most recent literature concerning domestication syndromes.

We want to thank reviewer 1 for their feedback and for agreeing to review this second part of the manuscript!

Compared to the previous version, the introduction part contains a little more information and the number of studied animals changed from 103 to 102.

Yes, the number of animals studied was updated. The analysis always ran on 102, so the results did not change. That was the first authors (my) mistake. I apologize, I mistyped the number while writing the introduction and methods and only noticed after the first submission. It is corrected now.

Overall, I consider the study as very important and worth publishing, however, I believe that the manuscript may benefit from restructuralization and reduction of the discussion. Some parts of the discussion are wordy and hard to follow. Some parts would better fit to the introduction, some parts are repetitions of the introduction or results.

We restructured the discussion and reduced it to the more relevant aspects of this paper.

Reviewer: 2

This is a timely and important contribution to domestication studies, not least given the recent vibrancy and frequency of papers surrounding domestication studies writ large. This is Greger Larson, I wanted to sign this review, since I am fascinated by this study and, in this case, it made more sense to me not to have this review be anonymous. I have a few comments that I'm hoping will improve this important manuscript.

Dear Professor Larson, thank you very much for agreeing to review this second part of the manuscript. Your kind words absolutely made the first author's (my) day! Thank you for providing such detailed comments and feedback on the first part of this manuscript and for doing so again here!

The abstract of any paper should summarise the entire manuscript. The authors set up the story nicely, but then stop before presenting any results. This is not a movie preview, it is a paragraph that encapsulates the premise, the results, and the conclusions and I would ask the authors to rewrite the abstract to include every aspect of this paper. What did you find? Tell us!

*We now updated the abstract to include and encapsulate the entirety of the paper including the results. Line 22: "Our data indicate that domestic cats indeed, have smaller cranial volumes (implying smaller brains) relative to both, European wildcats (*Felis silvestris*) and the wild ancestors of domestic cats, the African wildcats (*F. lybica*), verifying older results. We further found that..."*

The field is moving quickly and there has been another challenge to the neural crest hypothesis: DOI: 10.1093/genetics/iyab097. Wilkins has replied to this as well, and these latest entries in the debate should be included here. Having said that, I'm not sure how relevant the neural crest hypothesis is, or rather, it's important, but I would like to see the authors reference earlier papers that simply note, suggest, or observe smaller cranial volumes in domestic taxa (relative to their wild counterparts). Some of these references appear in the second paragraph, but they should be moved up since though Wilkins attempted to provide an explanation for the phenomenon, the observations have been around for decades prior to the NCT. This paper tests the observation, not the mechanism for its appearance, and that should be made clear by restricting the introduction.

We added this new reference regarding the NCC hypothesis by Johnsson, Henriksen and Wright to the introduction in line 55: "However, this downregulation may also cause correlated changes to morphology, stress response and brain size (Wilkins, Wrangham and Fitch, 2014; Wilkins, 2017). Although this hypothesis has found a lot of research support, others are critical of it (Johnsson, Henriksen and Wright, 2021)." To address these points raised we also added references purely observing changes to the cranial volume above our introduction of the NCC hypothesis in line 46: "In particular, changes to cranial volume have been well documented across species, including sheep, rabbits, dogs and many more (Darwin, 1868; Dieter Kruska, 1988; Wright, 2015)."

The authors mention other taxa in the abstract, but these seem to be missing in the introduction. The Lord et al. response in TREE regarding skull size is a good place to find and include some of these, and there are recent papers by Marcelo Sanchez that would fit well here too.

We now include species examples in the introduction in line 48: "In particular, changes to cranial volume have been well documented across species, including sheep, rabbits, dogs and many more (Darwin, 1868; Dieter Kruska, 1988; Wright, 2015)."

For the paragraph starting on line 65, how did these authors define 'feral' and 'domestic' cats. What criteria did they use, how many samples did they test, and from what geographical locations?

*We go into more detail regarding this issue in the methods in line 122: "It is unclear if the feral cats Hemmer presented in his study were indeed feral domestic cats or hybrids of European wildcats and domestic cats. The majority of Hemmer's feral domestic cat data has its origin in Klatt (1912), who mentioned the possibility of these individuals being European wildcat/domestic hybrids. Therefore, we used skulls of known European wildcat x domestic cat hybrids..." and the discussion in line 323: "Hemmer's data for feral cats originated from Klatt (1912), who mentioned the possibility of these individuals being hybrid offspring of *F. silvestris* and *F. catus*." Summed up: Hemmer*

reused data published by Klatt in 1912 who simply argues that he thinks those cats are feral. This was the reason why we chose to use hybrid cats in our data collection because based on the time those data were collected those cats very likely could have been hybrids.

Line 77. What species is the Sardinian wild cat?

The Sardinian wildcat is a Felis lybica cat. We now clarify this issue in line 86: “except Groves (1989), who presented data on cranial indices of Sardinian wildcats (F. lybica) to demonstrate that they had larger cranial volumes than those of domestic cats and to support their recognition as F. lybica.”

Line 107. How was it determined that the hybrids were in fact hybrids? Were they F1 hybrids, or were they thought to have only a small fraction of one lineage in their ancestry, and how was this known? The references on line 117 are great, but as a reader I’d prefer not to have to look these up. Can you provide a summary of how this determination was made here?

We now provide more information regarding the hybridisation and determination of hybrids in line 153: “In brief, genetic analysis involved a panel of 35 SNPs which had been developed to distinguish wildcats from domestic cats. Cats were identified as wildcat with a LBQ score of 0.75 (Senn et al. 2019). Morphological analysis was based on seven key pelage characters with a minimum score of 19 out of 21 for wildcats (Kitchener et al. 2005). In addition, five skull characters were scored (Kitchener et al. 2005). Most of the hybrids originated from the hybrid swarm in Scotland and show varying levels of introgression between the parent species based on genetic and morphological data. Where there was a discrepancy between genetic and morphological evaluations, the default was to record these cats as hybrids. Where genetic data were not available, morphological characters from skins and skulls were used to classify wildcats, domestic cats and their hybrids.” We don’t have any information though regarding the exact generation of hybrids.

Line 120. This paragraph makes it sound like cranial volume was not measured, but later it’s clear that it was. Can you clarify that here? And has the glass bead methodology been used before? Can you cite previous papers where this method has been tested and the figures regarding the weight and volume measurements were ascertained?

We now clarify the issue regarding measurement of cranial volume by adding information in line 132: “Since the most complete comparison across species was presented over basal skull length by Hemmer, we decided to use basal skull measurements combined with cranial volume measurements for our own data collection in this replication study.”

We also include citations regarding the use of glass beads in previous work in line 167: “This methodology is well established in comparative research and has previously been used in cats (Yamaguchi, Driscoll, et al., 2004; Yamaguchi, Kitchener, et al., 2004).”

We further clarify how the conversion and weight of the glass beads was established in line 170: “To convert the glass bead weight measurements into volumes, we established a conversion factor of 10ml of glass beads weighing 15.37g prior to measuring the cranial volumes. Subsequently we were able to estimate cranial volumes from the weights of the beads..”

Line 184. It would be great to include the basic quantitative results here. I know the results are in the table, but it would be great to include them in the prose as well. Mean, standard error, and statistical significance testing between the groups.

We included the model estimates with the standard errors in the prose in line 210 :” Felis silvestris cranial volumes were the largest (estimate \pm standard error: 12 ± 1), and F. lybica cranial volumes were larger (estimate \pm standard error: 2.5 ± 1) than those of domestic cats, but smaller than those of F. silvestris. Hybrid cats’ (F. silvestris x F. catus) cranial volumes cluster between the two parent species (estimate \pm standard error: 5.6 ± 0.9).” We tried including significance levels as well but refrained from doing so in the end since it made the text too bulky to comfortably read.

We also included estimates and standard errors to the palate lengths in line 218: “Contrary to our hypothesis the results indicate that domestic cats) have longer palates compared to their

ancestral species *F. lybica* (estimate \pm standard error: -0.8 ± 0.3). *Felis silvestris* (estimate \pm standard error: 1 ± 0.3) has longer palates than those of domestic cats, which have similar palate lengths to those of hybrid cats.”

Line 225. Did the authors check if body size differences were also correlated with the cranial volume measurements. Could this be allometry?

Yes, we did check for that. In order to support this statement we ran a full/null model comparison. The null model only includes the proxy for body size as a predictor of the data distribution, whereas the full model includes both body size proxy and species. Since the full model was significantly better than the null model we can say that species is relevant in explaining the data distribution, but body size is still a significant factor within the model. Therefore we can say that while there is an allometric effect we also find a species specific effect. So one could argue that there is a species specific allometric effect.

Line 232. I wouldn't go so far as to say a growing body of evidence “supports” the NCT. It's more like their data could be consistent with it, but that's a completely different association. The review by Dominic Wright points this out and many other things besides.

*We changed the phrasing from “supports” to “is in line with”; see line 253: “While a growing body of research (e.g., red junglefowl *Gallus gallus* (Agnvall et al., 2018), village dogs *Canis familiaris* (Pendleton et al., 2018), and foxes *Vulpes vulpes* (Trut, Oskina and Kharlamova, 2009)) is in line with the neural crest cell hypothesis, the hypothesis has been challenged by others (Hansen Wheat, der Bijl and Wheat, 2020; Lord et al., 2020; Lord, Larson and Karlsson, 2020).”*

Line 235. This opening line is unnecessary and I would delete it since the paragraph words perfectly well without it. The authors also only discuss the NCT. There are other mechanisms that have been published, including the thyroid hormone, to explain the domestication syndrome, and it might be worth mentioning them here as well.

We deleted the first sentence of this paragraph as suggested and added a new paragraph discussing the thyroid hormone hypothesis starting line 293: “Another hypothesis with potential relevance to morphological changes in domestication research is the thyroid hormone hypothesis; this hypothesis was proposed by Crockford (2004) and was named and further discussed by Wilkins (2017). This hypothesis [...]”

Line 265. There is a new paper about the archaeologically determined time frame for the arrival of dingoes by Sue O'Connor that could be cited here. It would give some grounding to the phrase thousands of years ago.

Thank you for that suggestion, we added the citation in line 324 :” Regarding brain size and feralisation, research on dingoes, which are domestic dogs in Australia that became feral thousands of years ago (Balme, O'Connor and Fallon, 2018) .[...]”

Line 303. The point that Lord was making is that all of the foxes came from the same population that had been through two big bottlenecks before arriving in Russia. The argument in the paper is that there's no way to distinguish the appearance of these traits as a result of either selection or drift, since the population started with a seriously limited degree of genomic variation.

Thank you for the clarification. We adjusted the text accordingly in line 271: “Given the partial reliance of the neural crest cell hypothesis on the Belyaev data, Lord and colleagues argued that it is impossible to distinguish between these changes having emerged due to selection or genetic drift (Lord et al., 2020).”

Line 307. To say “Furthermore, the degree to which the Canadian farmed foxes showed domesticated behaviours was overstated by Lord and colleagues.” Without any supporting evidence or a citation is an empty accusation. If this was an overstatement, you have to say why you are saying that on the basis of what evidence. Scientific claims need to be supported by evidence, and not simply stated as fact without any empirical or observational evidence.

We deleted this paragraph.

Line 310. This is not the case. The fixation of these traits could easily result from drift, especially given the winnowing of the genomic variation through successive rounds of selection for tameness. What I'm not understanding is why this is relevant. This study is a test of the differences in cranial volume, not a defence of the NCT as a mechanism to explain the domestication syndrome, which again, the very appearance of which has been challenged by numerous studies cited in this paper (e.g. Hansen Wheat, etc.)

We also deleted this paragraph. It is indeed not relevant to the discussion of this data.

And with regard to that issue, showing a difference in the skull volumes of domestic and wild individuals is fascinating. But it does not pin point when, during the process of domestication, this took place. There is not reason that the drop in skull volume could not have occurred very recently, especially if all the domestic cats are house cats. Just something to consider.

First author opinion: I agree that there is no way of pinpointing exactly when this reduction happened. It will also depend on what we define as "recent". My (first authors) main concern in this is to make sure that differences we are picking up are not a side effect of line/pedigree breeding. I think that domestication and artificial selection (in the sense of creation of breeds like Persian cats etc.) are two very different things. This would be an interesting conversation to be had over a cup of coffee.

Line 363. The authors refer to the "neural crest/domestication syndrome hypothesis" but this conflates two very different things. The "domestication syndrome" is a suggestion that many different domestic animals possess a suite of similarly divergent characteristics relative to their individual wild ancestors. The NCT assumed the domestication syndrome to be an accurate observation, and is a proposed hypothesis to explain the appearance of those traits. There are hypotheses other than the NCT that have been discussed in the literature (e.g. DOI 10.18699/VJ17.262) as explanations for the domestication syndrome, recent studies have challenged the NCT (the Wright paper above) and Lord et al. challenged the existence of the domestication syndrome itself.

We updated the terminology accordingly and changed "neural crest/domestication syndrome hypothesis" to "neural crest cell hypothesis" throughout. We further include additional explanations of the domestication syndrome in line 293: "Another hypothesis with potential relevance to morphological changes in domestication research is the thyroid hormone hypothesis; this hypothesis was proposed by Crockford (2004) and was named and further discussed by Wilkins (2017). This hypothesis[...]"

I encourage the authors to clearly distinguish all these terms and debates and to be explicit about what this study is doing, and what can be gleaned from their results. It's a fantastic paper and with some tweaking and clarifying it'll be an excellent contribution to the literature.

We now provide clearer distinctions and expanded on other potential explanations regarding the domestication syndrome. We hope that our clarifications and efforts do be as explicit and clear in our terminology helped improve on this important point. We thank Professor Larson for all his input and detailed comments! Your efforts helped improve the manuscript and are very much appreciated! Also thank you for the praise, you made the first author's (my) day!